# On the State of the Art of Evaluation in Neural Language Models

**Gábor Melis**[†], **Chris Dyer**[†], **Phil Blunsom**[†‡]
{melisgl,cdyer,pblunsom}@google.com
[†]DeepMind
[‡]University of Oxford

## Abstract

Ongoing innovations in recurrent neural network architectures have provided a steady influx of apparently state-of-the-art results on language modelling benchmarks. However, these have been evaluated using differing codebases and limited computational resources, which represent uncontrolled sources of experimental variation. We reevaluate several popular architectures and regularisation methods with large-scale automatic black-box hyperparameter tuning and arrive at the somewhat surprising conclusion that standard LSTM architectures, when properly regularised, outperform more recent models. We establish a new state of the art on the Penn Treebank and Wikitext-2 corpora, as well as strong baselines on the Hutter Prize dataset.

## 1 Introduction

The scientific process by which the deep learning research community operates is guided by empirical studies that evaluate the relative quality of models. Complicating matters, the measured performance of a model depends not only on its architecture (and data), but it can strongly depend on hyperparameter values that affect learning, regularisation, and capacity. This hyperparameter dependence is an often inadequately controlled source of variation in experiments, which creates a risk that empirically unsound claims will be reported.

In this paper, we use a black-box hyperparameter optimisation technique to control for hyperparameter effects while comparing the relative performance of language modelling architectures based on LSTMs, Recurrent Highway Networks (Zilly et al., 2016) and NAS (Zoph & Le, 2016). We specify flexible, parameterised model families with the ability to adjust embedding and recurrent cell sizes for a given parameter budget and with fine grain control over regularisation and learning hyperparameters.

Once hyperparameters have been properly controlled for, we find that LSTMs outperform the more recent models, contra the published claims. Our result is therefore a demonstration that replication failures can happen due to poorly controlled hyperparameter variation, and this paper joins other recent papers in warning of the under-acknowledged existence of replication failure in deep learning (Henderson et al., 2017; Reimers & Gurevych, 2017). However, we do show that careful controls are possible, albeit at considerable computational cost.

Several remarks can be made in light of these results. First, as (conditional) language models serve as the central building block of many tasks, including machine translation, there is little reason to expect that the problem of unreliable evaluation is unique to the tasks discussed here. However, in machine translation, carefully controlling for hyperparameter effects would be substantially more expensive because standard datasets are much larger. Second, the research community should strive for more consensus about appropriate experimental methodology that balances costs of careful experimentation with the risks associated with false claims. Finally, more attention should be paid to hyperparameter sensitivity. Models that introduce many new hyperparameters or which perform well only in narrow ranges of hyperparameter settings should be identified as such as part of standard publication practice.

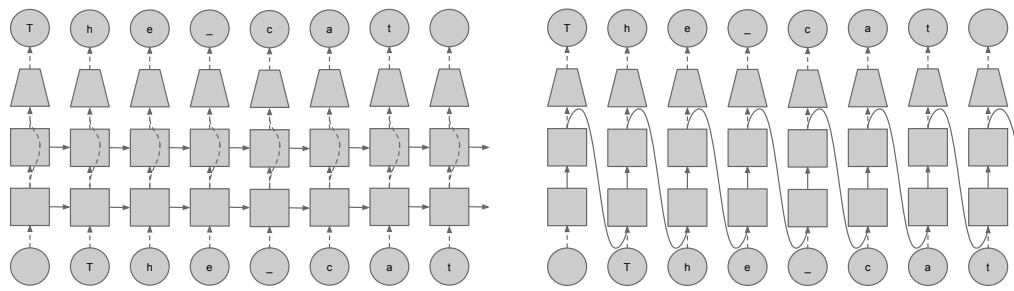

(a) two-layer LSTM/NAS with skip connections    (b) RHN with two processing steps per input

Figure 1: Recurrent networks with optional down-projection (trapezoids), per-step and per-sequence dropout (dashed and solid lines).

## 2 MODELS

Our focus is on three recurrent architectures:

- The Long Short-Term Memory (Hochreiter & Schmidhuber, 1997) serves as a well known and frequently used baseline.

- The recently proposed Recurrent Highway Network (Zilly et al., 2016) is chosen because it has demonstrated state-of-the-art performance on a number of datasets.

- Finally, we also include NAS (Zoph & Le, 2016), because of its impressive performance and because its architecture was the result of an automated reinforcement learning based optimisation process.

Our aim is strictly to do better model comparisons for these architectures and we thus refrain from including techniques that are known to push perplexities even lower, but which are believed to be largely orthogonal to the question of the relative merits of these recurrent cells. In parallel work with a remarkable overlap with ours, Merity et al. (2017) demonstrate the utility of adding a Neural Cache (Grave et al., 2016). Building on their work, Krause et al. (2017) show that Dynamic Evaluation (Graves, 2013) contributes similarly to the final perplexity.

As pictured in Fig. 1a, our models with LSTM or NAS cells have all the standard components: an input embedding lookup table, recurrent cells stacked as layers with additive skip connections combining outputs of all layers to ease optimisation. There is an optional down-projection whose presence is governed by a hyperparameter from this combined output to a smaller space which reduces the number of output embedding parameters. Unless otherwise noted, input and output embeddings are shared, see (Inan et al., 2016) and (Press & Wolf, 2016).

Dropout is applied to feedforward connections denoted by dashed arrows in the figure. From the bottom up: to embedded inputs (*input dropout*), to connections between layers (*intra-layer dropout*), to the combined and the down-projected outputs (*output dropout*). All these dropouts have random masks drawn independently per time step, in contrast to the dropout on recurrent states where the same mask is used for all time steps in the sequence.

RHN based models are typically conceived of as a single horizontal "highway" to emphasise how the recurrent state is processed through time. In Fig. 1b, we choose to draw their schema in a way that makes the differences from LSTMs immediately apparent. In a nutshell, the RHN state is passed from the topmost layer to the lowest layer of the next time step. In contrast, each LSTM layer has its own recurrent connection and state.

The same dropout variants are applied to all three model types, with the exception of intra-layer dropout which does not apply to RHNs since only the recurrent state is passed between the layers.

For the recurrent states, all architectures use either variational dropout (Gal & Ghahramani, 2016, *state dropout*)[1] or recurrent dropout (Semeniuta et al., 2016), unless explicitly noted otherwise.

# 3 EXPERIMENTAL SETUP

## 3.1 DATASETS

We compare models on three datasets. The smallest of them is the Penn Treebank corpus by Marcus et al. (1993) with preprocessing from Mikolov et al. (2010). We also include another word level corpus: Wikitext-2 by Merity et al. (2016). It is about twice the size of Penn Treebank with a larger vocabulary and much lighter preprocessing. The third corpus is Enwik8 from the Hutter Prize dataset (Hutter, 2012). Following common practice, we use the first 90 million characters for training, and the remaining 10 million evenly split between validation and test.

# 4 TRAINING DETAILS

When training word level models we follow common practice and use a batch size of 64, truncated backpropagation with 35 time steps, and we feed the final states from the previous batch as the initial state of the subsequent one. At the beginning of training and test time, the model starts with a zero state. To bias the model towards being able to easily start from such a state at test time, during training, with probability 0.01 a constant zero state is provided as the initial state.

Optimisation is performed by Adam (Kingma & Ba, 2014) with $\beta_1 = 0$ but otherwise default parameters ($\beta_2 = 0.999$, $\epsilon = 10^{-9}$). Setting $\beta_1$ so turns off the exponential moving average for the estimates of the means of the gradients and brings Adam very close to RMSProp without momentum, but due to Adam's bias correction, larger learning rates can be used.

Batch size is set to 64. The learning rate is multiplied by 0.1 whenever validation performance does not improve ever during 30 consecutive checkpoints. These checkpoints are performed after every 100 and 200 optimization steps for Penn Treebank and Wikitext-2, respectively.

For character level models (i.e. Enwik8), the differences are: truncated backpropagation is performed with 50 time steps. Adam's parameters are $\beta_2 = 0.99$, $\epsilon = 10^{-5}$. Batch size is 128. Checkpoints are only every 400 optimisation steps and embeddings are not shared.

# 5 EVALUATION

For evaluation, the checkpoint with the best validation perplexity found by the tuner is loaded and the model is applied to the test set with a batch size of 1. For the word based datasets, using the training batch size makes results worse by 0.3 PPL while Enwik8 is practically unaffected due to its evaluation and training sets being much larger. Preliminary experiments indicate that MC averaging would bring a small improvement of about 0.4 in perplexity and 0.005 in bits per character, similar to the results of Gal & Ghahramani (2016), while being a 1000 times more expensive which is prohibitive on larger datasets. Therefore, throughout we use the mean-field approximation for dropout at test time.

## 5.1 HYPERPARAMETER TUNING

Hyperparameters are optimised by Google Vizier (Golovin et al., 2017), a black-box hyperparameter tuner based on batched GP bandits using the expected improvement acquisition function (Desautels et al., 2014). Tuners of this nature are generally more efficient than grid search when the number of hyperparameters is small. To keep the problem tractable, we restrict the set of hyperparameters to *learning rate*, *input embedding ratio*, *input dropout*, *state dropout*, *output dropout*, *weight decay*. For deep LSTMs, there is an extra hyperparameter to tune: *intra-layer dropout*. Even with this small set, thousands of evaluations are required to reach convergence.

---

[1] Of the two parameterisations, we used the one in which there is further sharing of masks between gates rather than independent noise for the gates.

| Model | Size | Depth | Valid | Test |
|---|---|---|---|---|
| Medium LSTM, Zaremba et al. (2014) | 10M | 2 | 86.2 | 82.7 |
| Large LSTM, Zaremba et al. (2014) | 24M | 2 | 82.2 | 78.4 |
| VD LSTM, Press & Wolf (2016) | 51M | 2 | 75.8 | 73.2 |
| VD LSTM, Inan et al. (2016) | 9M | 2 | 77.1 | 73.9 |
| VD LSTM, Inan et al. (2016) | 28M | 2 | 72.5 | 69.0 |
| VD RHN, Zilly et al. (2016) | 24M | 10 | 67.9 | 65.4 |
| NAS, Zoph & Le (2016) | 25M | - | - | 64.0 |
| NAS, Zoph & Le (2016) | 54M | - | - | 62.4 |
| AWD-LSTM, Merity et al. (2017) † | 24M | 3 | 60.0 | 57.3 |
| LSTM | | 1 | 61.8 | 59.6 |
| LSTM | | 2 | 63.0 | 60.8 |
| LSTM | 10M | 4 | 62.4 | 60.1 |
| RHN | | 5 | 66.0 | 63.5 |
| NAS | | 1 | 65.6 | 62.7 |
| LSTM | | 1 | 61.4 | 59.5 |
| LSTM | | 2 | 62.1 | 59.6 |
| LSTM | 24M | 4 | 60.9 | 58.3 |
| RHN | | 5 | 64.8 | 62.2 |
| NAS | | 1 | 62.1 | 59.7 |

Table 1: Validation and test set perplexities on Penn Treebank for models with different numbers of parameters and depths. All results except those from Zaremba are with shared input and output embeddings. VD stands for Variational Dropout from Gal & Ghahramani (2016). †: parallel work.

**Parameter budget.**   Motivated by recent results from Collins et al. (2016), we compare models on the basis of the total number of trainable parameters as opposed to the number of hidden units. The tuner is given control over the presence and size of the down-projection, and thus over the tradeoff between the number of embedding vs. recurrent cell parameters. Consequently, the cells' hidden size and the embedding size is determined by the actual parameter budget, depth and the *input embedding ratio* hyperparameter.

For Enwik8 there are relatively few parameters in the embeddings since the vocabulary size is only 205. Here we choose not to share embeddings and to omit the down-projection unconditionally.

# 6  RESULTS

## 6.1  PENN TREEBANK

We tested LSTMs of various depths and an RHN of depth 5 with parameter budgets of 10 and 24 million matching the sizes of the Medium and Large LSTMs by (Zaremba et al., 2014). The results are summarised in Table 1.

Notably, in our experiments even the RHN with only 10M parameters has better perplexity than the 24M one in the original publication. Our 24M version improves on that further. However, a shallow LSTM-based model with only 10M parameters enjoys a very comfortable margin over that, with deeper models following near the estimated noise range. At 24M, all depths obtain very similar results, reaching 58.3 at depth 4. Unsurprisingly, NAS whose architecture was chosen based on its performance on this dataset does almost equally well, even better than in Zoph & Le (2016).

## 6.2  WIKITEXT-2

Wikitext-2 is not much larger than Penn Treebank, so it is not surprising that even models tuned for Penn Treebank perform reasonably on this dataset, and this is in fact how results in previous works were produced. For a fairer comparison, we also tune hyperparameters on the same dataset. In Table 2, we report numbers for both approaches. All our results are well below the previous state of the are for models without dynamic evaluation or caching. That said, our best result, 65.9 compares

| Model | Size | Depth | Valid | Test |
|---|---|---|---|---|
| VD LSTM, Merity et al. (2016) | 20M | 2 | 101.7 | 96.3 |
| VD+Zoneout LSTM, Merity et al. (2016) | 20M | 2 | 108.7 | 100.9 |
| VD LSTM, Inan et al. (2016) | 22M | 2 | 91.5 | 87.7 |
| AWD-LSTM, Merity et al. (2017) † | 33M | 3 | 68.6 | 65.8 |
| LSTM (tuned for PTB) | | 1 | 88.4 | 83.2 |
| LSTM | | 1 | 72.7 | 69.1 |
| LSTM | 10M | 2 | 73.8 | 70.7 |
| LSTM | | 4 | 78.3 | 74.3 |
| RHN | | 5 | 83.5 | 79.5 |
| NAS | | 1 | 79.6 | 75.9 |
| LSTM (tuned for PTB) | | 1 | 79.8 | 76.3 |
| LSTM | | 1 | 69.3 | 65.9 |
| LSTM | 24M | 2 | 69.1 | 65.9 |
| LSTM | | 4 | 70.5 | 67.6 |
| RHN | | 5 | 78.1 | 75.6 |
| NAS | | 1 | 73.0 | 69.8 |

Table 2: Validation and test set perplexities on Wikitext-2. All results are with shared input and output embeddings. †: parallel work.

favourably even to the Neural Cache (Grave et al., 2016) whose innovations are fairly orthogonal to the base model.

Shallow LSTMs do especially well here. Deeper models have gradually degrading perplexity, with RHNs lagging all of them by a significant margin. NAS is not quite up there with the LSTM suggesting its architecture might have overfitted to Penn Treebank, but data for deeper variants would be necessary to draw this conclusion.

### 6.3 ENWIK8

In contrast to the previous datasets, our numbers on this task (reported in BPC, following convetion) are slightly off the state of the art. This is most likely due to optimisation being limited to 14 epochs which is about a tenth of what the model of Zilly et al. (2016) was trained for. Nevertheless, we match their smaller RHN with our models which are very close to each other. NAS lags the other models by a surprising margin at this task.

## 7 ANALYSIS

On two of the three datasets, we improved previous results substantially by careful model specification and hyperparameter optimisation, but the improvement for RHNs is much smaller compared to that for LSTMs. While it cannot be ruled out that our particular setup somehow favours LSTMs, we believe it is more likely that this effect arises due to the original RHN experimental condition having been tuned more extensively (this is nearly unavoidable during model development).

Naturally, NAS benefitted only to a limited degree from our tuning, since the numbers of Zoph & Le (2016) were already produced by employing similar regularisation methods and a grid search. The small edge can be attributed to the suboptimality of grid search (see Section 7.3).

In summary, the three recurrent cell architectures are closely matched on all three datasets, with minuscule differences on Enwik8 where regularisation matters the least. These results support the claims of Collins et al. (2016), that capacities of various cells are very similar and their apparent differences result from trainability and regularisation. While comparing three similar architectures cannot prove this point, the inclusion of NAS certainly gives it more credence. This way we have two of the best human designed and one machine optimised cell that was the top performer among thousands of candidates.

| Model | Size | Depth | Valid | Test |
|---|---|---|---|---|
| Stacked LSTM, Graves (2013) | 21M | 7 | - | 1.67 |
| Grid LSTM, Kalchbrenner et al. (2015) | 17M | 6 | - | 1.47 |
| MI-LSTM, Wu et al. (2016) | 17M | 1 | - | 1.44 |
| LN HM-LSTM, Chung et al. (2016) | 35M | 3 | - | 1.32 |
| ByteNet, Kalchbrenner et al. (2016) | - | 25 | - | 1.31 |
| VD RHN, Zilly et al. (2016) | 23M | 5 | - | 1.31 |
| VD RHN, Zilly et al. (2016) | 21M | 10 | - | 1.30 |
| VD RHN, Zilly et al. (2016) | 46M | 10 | - | 1.27 |
| LSTM | | 4 | 1.29 | 1.31 |
| RHN | 27M | 5 | 1.30 | 1.31 |
| NAS | | 4 | 1.38 | 1.40 |
| LSTM | | 4 | 1.28 | 1.30 |
| RHN | 46M | 5 | 1.29 | 1.30 |
| NAS | | 4 | 1.32 | 1.33 |

Table 3: Validation and test set BPCs on Enwik8 from the Hutter Prize dataset.

## 7.1 THE EFFECT OF INDIVIDUAL FEATURES

Down-projection was found to be very beneficial by the tuner for some depth/budget combinations. On Penn Treebank, it improved results by about 2–5 perplexity points at depths 1 and 2 at 10M, and depth 1 at 24M, possibly by equipping the recurrent cells with more capacity. The very same models benefited from down-projection on Wikitext-2, but even more so with gaps of about 10–18 points which is readily explained by the larger vocabulary size.

We further measured the contribution of other features of the models in a series of experiments. See Table 4. To limit the number of resource used, in these experiments only individual features were evaluated (not their combinations) on Penn Treebank at the best depth for each architecture (LSTM or RHN) and parameter budget (10M or 24M) as determined above.

First, we untied input and output embeddings which made perplexities worse by about 6 points across the board which is consistent with the results of Inan et al. (2016).

Second, without variational dropout the RHN models suffer quite a bit since there remains no dropout at all in between the layers. The deep LSTM also sees a similar loss of perplexity as having intra-layer dropout does not in itself provide enough regularisation.

Third, we were also interested in how recurrent dropout (Semeniuta et al., 2016) would perform in lieu of variational dropout. Dropout masks were shared between time steps in both methods, and our results indicate no consistent advantage to either of them.

## 7.2 MODEL SELECTION

With a large number of hyperparameter combinations evaluated, the question of how much the tuner overfits arises. There are multiple sources of noise in play,

(a) non-deterministic ordering of floating-point operations in optimised linear algebra routines,
(b) different initialisation seeds,
(c) the validation and test sets being finite samples from a infinite population.

To assess the severity of these issues, we conducted the following experiment: models with the best hyperparameter settings for Penn Treebank and Wikitext-2 were retrained from scratch with various initialisation seeds and the validation and test scores were recorded. If during tuning, a model just got a lucky run due to a combination of (a) and (b), then retraining with the same hyperparameters but with different seeds would fail to reproduce the same good results.

There are a few notable things about the results. First, in our environment (Tensorflow with a single GPU) even with the same seed as the one used by the tuner, the effect of (a) is almost as large as that of (a) and (b) combined. Second, the variance induced by (a) and (b) together is roughly equivalent to an absolute difference of 0.4 in perplexity on Penn Treebank and 0.5 on Wikitext-2.

| Model | Size 10M | | | Size 24M | | |
|---|---|---|---|---|---|---|
| | Depth | Valid | Test | Depth | Valid | Test |
| LSTM | 1 | 61.8 | 59.6 | 4 | 60.9 | 58.3 |
| - Shared Embeddings | 1 | 67.6 | 65.2 | 4 | 65.6 | 63.2 |
| - Variational Dropout | 1 | 62.9 | 61.2 | 4 | 66.3 | 64.5 |
| + Recurrent Dropout | 1 | 62.8 | 60.6 | 4 | 65.2 | 62.9 |
| + Untied gates | 1 | 61.4 | 58.9 | 4 | 64.0 | 61.3 |
| + Tied gates | 1 | 61.7 | 59.6 | 4 | 60.4 | 58.0 |
| RHN | 5 | 66.0 | 63.5 | 5 | 64.8 | 62.2 |
| - Shared Embeddings | 5 | 72.3 | 69.5 | 5 | 67.4 | 64.6 |
| - Variational Dropout | 5 | 74.4 | 71.7 | 5 | 74.7 | 71.7 |
| + Recurrent Dropout | 5 | 65.5 | 63.0 | 5 | 63.4 | 61.0 |

Table 4: Validation and test set perplexities on Penn Treebank for variants of our best LSTM and RHN models of two sizes.

Third, the validation perplexities of the best checkpoints are about one standard deviation lower than the sample mean of the reruns, so the tuner could fit the noise only to a limited degree.

Because we treat our corpora as a single sequence, test set contents are not i.i.d., and we cannot apply techniques such as the bootstrap to assess (c). Instead, we looked at the gap between validation and test scores as a proxy and observed that it is very stable, contributing variance of 0.12–0.3 perplexity to the final results on Penn Treebank and Wikitext-2, respectively.

We have not explicitly dealt with the unknown uncertainty remaining in the Gaussian Process that may affect model comparisons, apart from running it until apparent convergence. All in all, our findings suggest that a gap in perplexity of 1.0 is a statistically robust difference between models trained in this way on these datasets. The distribution of results was approximately normal with roughly the same variance for all models, so we still report numbers in a tabular form instead of plotting the distribution of results, for example in a violin plot (Hintze & Nelson, 1998).

## 7.3 SENSITIVITY

To further verify that the best hyperparameter setting found by the tuner is not a fluke, we plotted the validation loss against the hyperparameter settings. Fig. 2 shows one such typical plot, for a 4-layer LSTM. We manually restricted the ranges around the best hyperparameter values to around 15–25% of the entire tuneable range, and observed that the vast majority of settings in that neighbourhood produced perplexities within 3.0 of the best value. Widening the ranges further leads to quickly deteriorating results.

Satisfied that the hyperparameter surface is well behaved, we considered whether the same results could have possibly been achieved with a simple grid search. Omitting *input embedding ratio* because the tuner found having a down-projection suboptimal almost non-conditionally for this 4-layer LSTM, there remain six hyperparameters to tune. If there were 5 possible values on the grid for each hyperparameter (with one value in every 20% interval), then we would need $6^5$, nearly 8000 trials to get within 3.0 of the best perplexity achieved by the tuner in about 1500 trials.

## 7.4 TYING LSTM GATES

Normally, LSTMs have two independent gates controlling the retention of cell state and the admission of updates (Eq. 1). A minor variant which reduces the number of parameters at the loss of some flexibility is to tie the input and forget gates as in Eq. 2. A possible middle ground that keeps the number of parameters the same but ensures that values of the cell state $c$ remain in $[-1, 1]$ is to cap

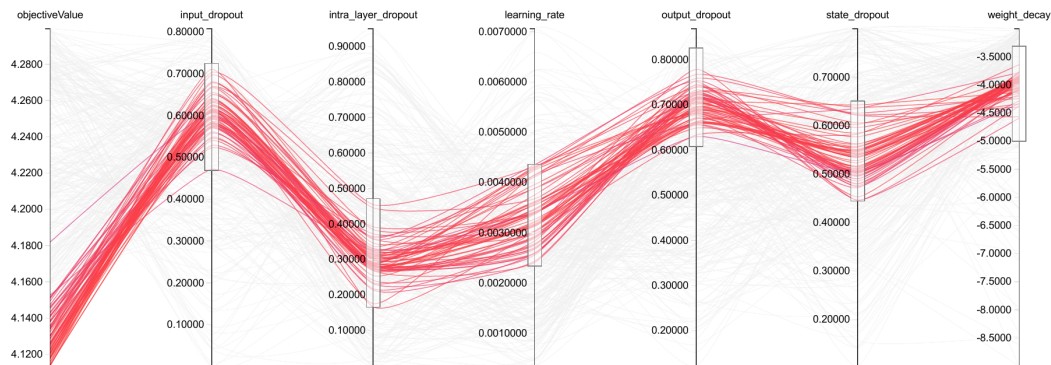

Figure 2: Average per-word negative log-likelihoods of hyperparameter combinations in the neighbourhood of the best solution for a 4-layer LSTM with 24M weights on the Penn Treebank dataset.

the input gate as in Eq. 3.

$$\mathbf{c}_t = \mathbf{f}_t \odot \mathbf{c}_{t-1} + \mathbf{i}_t \odot \mathbf{j}_t \tag{1}$$

$$\mathbf{c}_t = \mathbf{f}_t \odot \mathbf{c}_{t-1} + (1 - \mathbf{f}_t) \odot \mathbf{j}_t \tag{2}$$

$$\mathbf{c}_t = \mathbf{f}_t \odot \mathbf{c}_{t-1} + \min(1 - \mathbf{f}_t, \mathbf{i}_t) \odot \mathbf{j}_t \tag{3}$$

Where the equations are based on the formulation of Sak et al. (2014). All LSTM models in this paper use the third variant, except those titled "Untied gates" and "Tied gates" in Table 4 corresponding to Eq. 1 and 2, respectively.

The results show that LSTMs are insensitive to these changes and the results vary only slightly even though more hidden units are allocated to the tied version to fill its parameter budget. Finally, the numbers suggest that deep LSTMs benefit from bounded cell states.

## 8   CONCLUSION

During the transitional period when deep neural language models began to supplant their shallower predecessors, effect sizes tended to be large, and robust conclusions about the value of the modelling innovations could be made, even in the presence of poorly controlled "hyperparameter noise." However, now that the neural revolution is in full swing, researchers must often compare competing deep architectures. In this regime, effect sizes tend to be much smaller, and more methodological care is required to produce reliable results. Furthermore, with so much work carried out in parallel by a growing research community, the costs of faulty conclusions are increased.

Although we can draw attention to this problem, this paper does not offer a practical methodological solution beyond establishing reliable baselines that can be the benchmarks for subsequent work. Still, we demonstrate how, with a huge amount of computation, noise levels of various origins can be carefully estimated and models meaningfully compared. This apparent tradeoff between the amount of computation and the reliability of results seems to lie at the heart of the matter. Solutions to the methodological challenges must therefore make model evaluation cheaper by, for instance, reducing the number of hyperparameters and the sensitivity of models to them, employing better hyperparameter optimisation strategies, or by defining "leagues" with predefined computational budgets for a single model representing different points on the tradeoff curve.

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
