# OpenReview forum: "On the State of the Art of Evaluation in Neural Language Models"
_ICLR.cc/2018/Conference — Accept (Poster)_

### Official Review · AnonReviewer1 · 2017-11-24
**a useful exercise**

**Rating:** 7
**Confidence:** 2

**Review:**

The submitted manuscript describes an exercise in performance comparison for neural language models under standardization of the hyperparameter tuning and model selection strategies and costs.  This type of study is important to give perspective to non-standardized performance scores reported across separate publications, and indeed the results here are interesting as they favour relatively simpler structures.

I have a favourable impression of this paper but would hope another reviewer is more familiar with the specific application domain than I am.

---

> ### Author Response · Authors · 2017-12-11
> **Re: a useful exercise**
>
> We thank AnonReviewer1 for their review.
>
> We would like to point out that the state-of-the-art results and model comparisons are only part of the message. More importantly, we argue that the way model evaluation is performed is often unsatisfactory. Evaluation at a single hyperparameter setting, failing to control for dominant sources of variation make results unreliable and slow down progress.

---

### Official Review · AnonReviewer3 · 2017-11-27
**With extensive tuning, LSTM beats other new models**

**Rating:** 5
**Confidence:** 5

**Review:**

The authors did extensive tuning of the parameters for several recurrent neural architectures. The results are interesting. However the corpus the authors choose are quite small, the variance of the estimate will be quite high, I suspect whether the same conclusions could be drawn.

It would be more convincing if there are experiments on the billion word corpus or other larger datasets, or at least on a corpus with 50 million tokens. This will use significant resources and is much more difficult, but it's also really valuable, because it's much more close to real world usage of language models. And less tuning is needed for these larger datasets.

Finally it's better to do some experiments on machine translation or speech recognition and see how the improvement on BLEU or WER could get.

---

> ### Public Comment · (anonymous) · 2017-12-02
> **Too harsh**
>
> PTB is very much a standard baseline in the space that people routinely publish perf-based papers on, so I think it's a little harsh to knock people for publishing on it, especially when they make the effort to get results on multiple larger datasets, such as Wikitext-2.
>
> This is not a paper about the utility of hyperparameter optimization, the fact that that works has been well established. It's a paper about how hyperparameter optimization wasn't properly used on a bunch of standard benchmarks in this space, which has already proven very valuable. I really don't think it's necessary to request results on MT or ASR

---

> ### Author Response · Authors · 2017-12-11
> **Re: With extensive tuning, LSTM beats other new models**
>
> We feel that AnonReviewer3 might have missed that the main message of the paper was that evaluation - as it's generally performed - is unreliable. Our results suggest that state-of-the-art results are only superficially considered, and variance and parameter sensitivity are likewise given short shrift.
>
> The main criticism seems to center on evaluating models on datasets that are too small which increases evaluation variance and the results are thus not trustworthy. That is a very good summary of the main message of the paper! We agree that small datasets are problematic, but one cannot refute previous results that were obtained on small datasets using large datasets. Furthermore, we do hyperparameter tuning and a careful analysis of the variance. Furthermore, the third dataset (enwik8) is a large character based corpus and we still improve previously reported LSTM results by a substantial margin.
>
> Finally, to do this kind of study we chose language modelling because of its relevance to all kinds recurrent neural models while being simpler than machine translation and speech recognition models. We have demonstrated evaluation problems in this simple and relevant setting. It is unclear why the reviewer requests results on MT and ASR.

---

### Official Review · AnonReviewer2 · 2017-11-30
**Important big-picture work in a fast-moving field**

**Rating:** 8
**Confidence:** 3

**Review:**

The authors perform a comprehensive validation of LSTM-based word and character language models, establishing that recent claims that other structures can consistently outperform the older stacked LSTM architecture result from failure to fully explore the hyperparameter space. Instead, with more thorough hyperparameter search, LSTMs are found to achieve state-of-the-art results on many of these language modeling tasks.
This is a significant result in language modeling and a milestone in deep learning reproducibility research. The paper is clearly motivated and authoritative in its conclusions but it's somewhat lacking in detailed model or experiment descriptions.

Some further points:

- There are several hyperparameters set to the "standard" or "default" value, like Adam's beta parameter and the batch size/BPTT length. Even if it would be prohibitive to include them in the overall hyperparameter search, the community is curious about their effect and it would be interesting to hear if the authors' experience suggests that these choices are indeed reasonably well-justified.

- The description of the model is ambiguous on at least two points. First, it wasn't completely clear to me what the down-projection is (if it's simply projecting down from the LSTM hidden size to the embedding size, it wouldn't represent a hyperparameter the tuner can set, so I'm assuming it's separate and prior to the conventional output projection). Second, the phrase "additive skip connections combining outputs of all layers" has a couple possible interpretations (e.g., skip connections that jump from each layer to the last layer or (my assumption) skip connections between every pair of layers?).

- Fully evaluating the "claims of Collins et al. (2016), that capacities of various cells are very similar and their apparent
differences result from trainability and regularisation" would likely involve adding a fourth cell to the hyperparameter sweep, one whose design is more arbitrary and is neither the result of human nor machine optimization.

- The reformulation of the problem of deciding embedding and hidden sizes into one of allocating a fixed parameter budget towards the embedding and recurrent layers represents a significant conceptual step forward in understanding the causes of variation in model performance.

- The plot in Figure 2 is clear and persuasive, but for reproducibility purposes it would also be nice to see an example set of strong hyperparameters in a table. The history of hyperparameter proposals and their perplexities would also make for a fantastic dataset for exploring the structure of RNN hyperparameter spaces. For instance, it would be helpful for future work to know which hyperparameters' effects are most nearly independent of other hyperparameters.

- The choice between tied and clipped (Sak et al., 2014) LSTM gates, and their comparison to standard untied LSTM gates, is discussed only minimally, although it represents a significant difference between this paper and the most "standard" or "conventional" LSTM implementation (e.g., as provided in optimized GPU libraries). In addition to further discussion on this point, this result also suggests evaluating other recently proposed "minor changes" to the LSTM architecture such as multiplicative LSTM (Krause et al., 2016)

- It would also have been nice to see a comparison between the variational/recurrent dropout parameterization "in which there is further sharing of masks between gates" and the one with "independent noise for the gates," as described in the footnote. There has been some confusion in the literature as to which of these parameterizations is better or more standard; simply justifying the choice of parameterization a little more would also help.

---

> ### Author Response · Authors · 2017-12-11
> **Re: Important big-picture work in a fast-moving field**
>
> We thank AnonReviewer2 for the thoughtful and detailed review, let us address the points brought up one by one in the original order (we will likewise clarify these points in the paper):
>
> - Some hyperparameters were indeed left at "default" values because our tuner cannot efficiently tune a large set of hyperparameters. Still we did tuning studies with lower and higher BPTT lengths, batch sizes and including Adam parameters (beta1, beta2, epsilon) and with other optimizers to make sure that our intuition about what hyperparameters are most important is correct. We did a tuning study with all hyperparameters (about 40 hyperparameters in total) to catch any unexpected parameter combinations even if it was a long shot due to the aforementioned tuner inefficiency.
>
> - Yes, the down-projection is simply projecting down from the LSTM hidden size to the embedding size. The ratio of the embedding size and cell size is a tuneable. The cell and embedding sizes are computed from the budget and this input_embedding_ratio hyperparameter. As the paper puts it: "The tuner is given control over the presence and size of the down-projection, and thus over the tradeoff between the number of embedding vs. recurrent cell parameters. Consequently, the cells’ hidden size and the embedding size is determined by the actual parameter budget, depth and the input embedding ratio hyperparameter."
>
> - Yes, we didn't find a very different cell with promising results in the literature.
>
> - No comment.
>
> - We are working on factoring out the code from a larger system and providing training scripts with the tuned hyperparameters.
>
> - The Multiplicative LSTM is indeed interesting. We did some preliminary investigation and could not make it perform very well. In the end, it was excluded to avoid adding further multipliers to our already very high resource consumption.
>
> - We used shared masks because of implementation convenience and for computational considerations.

---

### Public Comment · (anonymous) · 2017-11-27
**Whether to use Down-projection?**

I found two confusing statements from the paper:

Section 7.1:
Down-projection was found to be very beneficial by the tuner for some depth/budget combinations.
On Penn Treebank, it improved results by about 2–5 perplexity points at depths 1 and 2 at 10M, and
depth 1 at 24M

Section 7.3:
Omitting input embedding ratio because
the tuner found having a down-projection suboptimal almost non-conditionally for this model

I think the first one is suggesting down-projection to be beneficial and the second one is suggesting that tuner finds it suboptimal...

---

> ### Author Response · Authors · 2017-11-27
> **Re: Whether to use Down-projection?**
>
> Section 7.1 discusses the effect of down-projection in general for various models (depth/budget). Section 7.3 uses a 4-layer LSTM with 24M weights as an example, for which the down-projection is universally suboptimal.
>
> We agree that Section 7.3 is not very clear on this.

---

> > ### Public Comment · (anonymous) · 2017-11-29
> > **Re**
> >
> > 4-layer LSTM 24M is not the universal best setting for both PTB and Wikitext-2...so I guess the lesson to be learned here is that we need to consider down-projection as a hyper-parameter, and this bottleneck-structure may or may not be useful.

---

> > > ### Author Response · Authors · 2017-11-29
> > > **Re: Re**
> > >
> > > This is exactly what we did. The presence and size of the down-projection was a tuned hyperparameter. Section 7.1 discusses for which models it was useful and for which it wasn't.

---

### Public Comment · ~Slav_Petrov1 · 2017-11-29
**Question about Transfer Experiment**

Very nice paper!

I was wondering whether you could provide more details on the transfer experiment where you tuned the hyperparameters of the LSTM on the PTB and then used those parameters to train a model on Wikitext-2. Do the tuned hyperparameters differ in interesting ways? Are different parameters needed because the vocabulary size is different or because the corpus size is different? Are there parameters that give decent performance across both tasks? Could you tune on both tasks simultaneously?

---

> ### Author Response · Authors · 2017-11-30
> **Re: Question about Transfer Experiment**
>
> The hyperparameters differ only in boring ways: Wikitext-2 needs a bit less intra layer and state dropout. This is very likely to be due to corpus size. Down-projection sizes are also a bit different due to the vocabulary size mismatch (when there is a down-projection at all).
>
> I'm not sure there are hyperparameters that work well on both, but yes, we could tune for combined (in whatever way) performance on a number of datasets. By doing this, we could learn more about how hyperparameters are best specified so that they are reasonably independent from datasets and also from other hyperparameters.

---

### Public Comment · (anonymous) · 2017-12-04
**Reproducible? Hyper-parameters?**

The main contribution of this paper is to show with extensive hyper-parameter tuning, LSTM can achieve a state of the art result for language modeling. However, it seems that the authors didn't give the concrete hyper-parameters for reproducing their results. This paper is lack of technical novelty and totally of a experimental work; thus the experiment setting is very important. Only showing HP tuning can get state of the art results if not enough, since it is a common sense that hyper-parameter tuning can improve performance for any machine learning models. I think the paper should at least describe their hyper-parameters setting (for example, learning rate, hidden size, dropout ratio, weight decay, etc) for getting their results or releasing the code for the community if possible.

---

> ### Author Response · Authors · 2017-12-04
> **Re: Reproducible? Hyper-parameters?**
>
> Indeed we have been asked for the hyperparameter settings on numerous occasions. Originally, we did not provide these details as the main message of the paper was not about the state of the results but model evaluation, but there is another, more fundemental reason too: any single hyperparameter setting would make it easy to compare a derivative work to our well tuned baseline, but at best that could prove that the new model is better (it could never prove that it's worse). More, two new models each evaluated with those hyperparameters would still be incomparable.
>
> For these reasons, we think that presently there is no way around tuning, and there is limited utility in publishing hyperparameter settings.
>
> That said, we are working on factoring out the code from a larger system and providing training scripts with the tuned hyperparameters.

---

### Public Comment · ~Stephen_Merity1 · 2017-12-08
**Strong recommend accept**

I would strongly recommend this paper be accepted for publication. It tackles and uncovers many important discussions regarding our models, our datasets, their sensitivity to hyperparameters, and the process of thoroughly comparing models and searching for hyper parameters. This helps inform a broader discussion about how we can ensure that the scientific process for our field is best followed. Thoroughly analyzing and forcing a reconsideration of the impact of proposed RNN model architectures (LSTM, RHN, NASCell) on these tasks is worth the price of admission by itself, let alone the many other learnings and investigations.

Informal Rating: 8
Confidence: 5
(work directly in this field and have recreated many aspects of the results since publication)

= Hyper parameters =

I'll reply to an earlier comment (search for "we have been asked for the hyperparameter settings on numerous occasions") as I'm interested in continuing discussion regarding your hesitance to release hyper parameters. Overall I am glad that you decided that you will release training scripts with the tuned hyperparameters as I genuinely think this will benefit the community going forward.

= Down projection =

To clarify, is this a specific and separate layer (a dense layer that takes the output of the LSTM, h, and projects it from |h| to the embedding size |e|) or is the final LSTM accepting an input of size |h| and internally down projecting it to an output of size |e|? I imagine it would be the former given that your single layer LSTMs appear to still use down projection, hence having a larger |h| than |e|? Did you experiment with modifying the last LSTM layer's sizings (which may breaking your skip connections but would be more "parameter efficient")?

= Skip connections =

Did you investigate models that didn't use skip connections for the RNNs? We have found in our work that such skip connections did not appear to be required, especially for models that are under four or so layers (though we have trained 6 or 7 layer models too - it just gets finicky at that stage). You and your team might have insights on this that we do not?

---

> ### Author Response · Authors · 2017-12-11
> **Re: Strong recommend accept**
>
> Thank you for taking the time to write the review.
>
> The down-projection is indeed the former version: it projects the output of the top LSTM (plus skip connections) to output_embedding_size. We didn't try the suggested variant.
>
> Yes, depth 1 and 2 LSTMs did not need skip connections but depth 4 suffered without them according to preliminary experiments. Alas, we have no further insight on this.

---

### Public Comment · ~William_Chan1 · 2017-12-17
**Hyperparams**

I wish/encourage the authors would post the hyperparams, it would make reproducing the results much easier, especially for the academic community which may not have the resources to run full hyperparameter searches (even if the scripts are released).

---

### Author Response · Authors · 2017-12-18
**Uploaded revision 2**

Changelist:

- Better NAS results with more tuning on Wikitext-2 and Enwik8. The story is the same, still lagging other models.
- Tiny adjustments related to down-projections that hopefully clarify things.

---

### Decision · Program_Chairs · 2018-01-29
**ICLR 2018 Conference Acceptance Decision**

**Decision:**

Accept (Poster)

**Comment:**

this submission demonstrates an existing loop-hole (?) in rushing out new neural language models by carefully (and expensively) running hyperparameter tuning of baseline approaches. i feel this is an important contribution, but as pointed out by some reviewers, i would have liked to see whether the conclusion stands even with a more realistic data (as pointed out by some in the field quite harshly, perplexity on PTB should not be considered seriously, and i believe the same for the other two corpora used in this submission.) that said, it's an important paper in general which will work as an alarm to the current practice in the field, and i recommend it to be accepted.